# Clinical characteristic and outcomes of pregnant women with COVID-19: The PROUDEST prospective cohort study

Lizandra Paravidine Sasaki[1,2◉*], Geraldo Magela Fernandes[1◉],
Ângelo Pereira da Silva[2], Yacara Ribeiro Pereira[2,3], Aleida Oliveira de Carvalho[2],
Felipe Motta[1], David Alves de Araújo Junior[1], Maria Eduarda Canellas de Castro[1],
Gabriela Profírio Jardim-Santos[1], Heidi Luise Schulte[1], Clara Correia de Siracusa[4],
Isadora Pastrana Rabelo[5], Pedro Sadi Monteiro[3], Fabiola Cristina Ribeiro Zucchi[4],
Agenor de Castro Moreira dos Santos Junior[5], Ismael Artur Costa-Rocha[6],
Jordana Grazziela Alves Coelho-dos-Reis[7], Patricia Shu Kurizky[1], Rosana Tristão[4],
Dayde Lane Mendonça Da Silva[2], Otávio de Toledo Nóbrega[1],
Alexandre Anderson de Sousa Munhoz Soares[1], Cleandro Pires de Albuquerque[1,2],
Ciro Martins Gomes[1,2,4,8], Laila Salmen Espindola[1], Olindo Assis Martins-Filho[6‡*],
Alberto Moreno Zaconeta[4‡], Licia Maria Henrique da Mota[1,2,7‡*]

1 Programa de Pós-Graduação em Ciências Médicas, Universidade de Brasília (UnB), Brasília, Distrito Federal, Brazil, 2 Hospital Universitário de Brasília, Universidade de Brasília (UnB), Brasília, Distrito Federal, Brazil, 3 Programa de Pós-graduação em Ciências da Saúde, Universidade de Brasília (UnB), Brasília, Distrito Federal, Brazil, 4 Faculdade de Medicina da Universidade de Brasília (UnB), Brasília, Distrito Federal, Brazil, 5 Secretaria de Saúde do Distrito Federal (SES/DF), Brasília, Distrito Federal, Brazil, 6 Instituto René Rachou, Fundação Oswaldo Cruz (FIOCRUZ-Minas), Belo Horizonte, MG, Brazil, 7 Laboratório de Virologia Básica e Aplicada, Instituto de Ciências Biológicas, Universidade Federal de Minas Gerais, Belo Horizonte, Brazil, 8 Programa de Pós-Graduação em Patologia Molecular, Universidade de Brasília (UnB), Brasília, Distrito Federal, Brazil

◉ These authors contributed equally to this work.
‡ These authors share senior authorship on this work.
* lizandra78@gmail.com (LMPS); liciamota228@gmail.com (LMHdaM); oamfilho@gmail.com (OAM-F)

## Abstract

The present study intended to characterize the clinical features and outcomes of SARS-CoV-2 infection at distinct pregnancy trimesters. A total of 260 pregnant women with SARS-CoV-2 infection at any pregnancy trimester were enrolled in a prospective follow-up study. Clinical features were recorded between the SARS-CoV-2 infection diagnosis towards delivery and postpartum period. ANOVA and Chi-square/Fisher tests, Bi- and multivariate analyses were performed to verify the effect of predictors on disease outcome, adjusted for the pregestational variables. Data demonstrated that anosmia (64.6%), nasal congestion/discharge (61.5%), headache (60.8%), ageusia (58.5%) and myalgia (58.5%) were the most common symptoms observed amongst pregnant women with non-severe COVID-19. Fever (44.6%) and dyspnea (36.5%) were associated with higher disease severity. Gestational diabetes mellitus (35.8%), systemic arterial hypertension (18.1%), preterm delivery (11.5%) and superimposed preeclampsia (6.2%) were reported as adverse pregnancy outcomes amongst pregnant women with COVID-19. Parturients with acute COVID-19

**Data availability statement:** All relevant data are within the manuscript and its Supporting Information files.

**Funding:** The author(s) received no specific funding for this work.

**Competing interests:** The authors have declared that no competing interests exist.

and pregnant women infected at the 3rd trimester presented more severe or critical outcomes as compared to those infected at 2nd and 1st trimesters. Preterm labor (Odds Ratio = 3.64), acute fetal distress (Prevalence Ratio = 2.40) and Apgar 1st minute score ≤ 7 (Prevalence Ratio = 2.56) were adverse outcomes reported in parturients with acute COVID-19 and those with severe or critical outcomes. Together these findings demonstrated that SARS-CoV-2 infection during pregnancy was associated with relevant maternal and neonatal adverse outcomes. The understanding of the clinical and obstetric outcomes of COVID-19 during pregnancy can provide insights to establish the most suitable approach for clinical management of pregnant women.

## Introduction

Since the initial outbreak of Coronavirus Disease-2019 (COVID-19), caused by severe acute respiratory syndrome coronavirus type 2 (SARS-CoV-2), the numbers of confirmed cases and reports of mortality/morbidity have continuously increased worldwide. The World Health Organization (WHO) declared COVID-19 a pandemic, which has been lasting for more than 3 years [1,2].

Globally, there have been 777,368,929 confirmed COVID-19 cases, including 7,087,731 deaths, reported to the WHO until February 2nd, 2025 [3]. According to the Brazilian Obstetric Observatory, as of February 20th, 2025, amongst the 2,247,377 COVID-19 cases reported in Brazil, 25,327 cases were diagnosed in pregnant and postpartum women (1%) [4].

At the onset of the pandemic, the real impacts of COVID-19 on pregnancy, maternal and fetal health, puerperium, as well as the consequences on the long-term development of children born to women infected during pregnancy, were completely unknown [5,6]. As the COVID-19 pandemic progressed, pregnant women were considered at high-risk for complications and disease severity, similar to other coronavirus infections outbreaks [5,6].

Multicenter studies evaluating maternal and perinatal outcomes following SARS-CoV-2 infection during pregnancy and the postpartum period include the REBRACO Study Group [7], COVI-PREG Cohort [8], INTERCOVID Multinational Cohort Study [9] and others. Moreover, all the outcomes of SARS-CoV-2 infection during pregnancy and the postpartum period are still unclear.

In this context, we conducted a prospective cohort study including pregnant women infected by SARS-CoV-2 during any pregnancy trimester, referred as *PRegnancy OUtcome and child Development – Effects of SARS-CoV-2 infection Trial* (PROUDEST), which analyzed effective clinical and epidemiological data collection [10]. The present study was designed to provide multiple deliverables, including primary and secondary outcomes. The primary goal focused on describing the clinical characteristics and maternal and perinatal outcomes of women infected with SARS-CoV-2 during any pregnancy trimester. The secondary outcome aimed at providing supporting evidence regarding the associations between maternal and perinatal

outcomes with i) pregnancy trimester of SARS-CoV-2 infection diagnosis ii) the severity of COVID-19 during pregnancy or iii) acute or convalescent phase of COVID-19.

## Materials and methods

### Study design

The present study was carried out according to the ethical principles stated in the Helsinki Declaration involving research with human beings. The protocol was submitted and approved by the Institutional review board of the medical faculty of the Universidade de Brasília (CAAE 32359620.0.0000.5558) in May 2020, with additional approval by the Comissão Nacional de Ética em Pesquisa (CONEP) in October 2021. All participants have provided written informed consent prior inclusion in the study. The study was registered on the Brazilian clinical trials platform (Rede Brasileira de Ensaios Clínicos – REBEC; https://ensaiosclinicos.gov.br/rg/RBR-65qxs2) in September 2020. All relevant data are available in the Harvard Dataverse repository 2024, "Repli": https://doi.org/10.7910/DVN/GOHQUY.

This is a prospective cohort study in the Brazilian Federal District (Brazil) that included pregnant women aged ≥18 years infected by SARS-CoV-2 during any pregnancy trimester. The participants were recruited as non-probabilistic convenience sampling, from May 2020 to May 2021 and followed prospectively from July 2020 to December 2021. Recruitment was conducted primarily through digital channels, including institutional websites, social media platforms, and professional networks. Institutional dissemination: an invitation to participate in the study was posted on official websites and internal bulletins of participating academic and healthcare institutions of Brasília, Federal District, Brazil. Social media: we utilized platforms such as Instagram and Facebook to circulate the study invitation. Posts included a brief description of the study objectives, eligibility criteria, and a phone number to schedule the first clinical study visit. Professional and academic groups: we shared the study invitation in closed messaging groups and online communities (e.g., WhatsApp groups) that included students, healthcare trainees, and professionals, particularly those involved in university hospitals or public health services. Snowball technique: we also encouraged participants and contacts to share the invitation with their peers, which further expanded the reach of our sample. While we acknowledge the inherent limitations of this strategy, including the potential for selection bias and reduced generalizability, we adopted this approach to maximize reach during a period when in-person recruitment was not feasible due to the public health crisis context. To mitigate bias, we sought to diversify dissemination channels and target a broad range of participants from different regions, institutions, and backgrounds.

The study was performed at two public hospitals – the University Hospital of Brasília and the Asa Norte Regional Hospital, both considered public reference centers for COVID-19 in the Federal District of Brazil.

To reduce the risk of confounding bias for the clinical characteristics and outcomes, pregnant women vaccinated for COVID-19 before or during pregnancy were excluded from the study. Moreover, pregnant women with confirmed co-infections, such as toxoplasmosis, syphilis, rubella, herpes, Chagas disease, cytomegalovirus, Zika virus, or human immunodeficiency virus (HIV) were also excluded. In addition, smokers and women with excessive alcohol/illicit drug use were also excluded. Failure to comply with the study protocol or attend the scheduled prenatal visits before delivery (loss to follow-up) were also exclusion criteria. Fig 1 provides a summary flowchart of study population recruitment and exclusion criteria.

The diagnosis of SARS-CoV-2 infection during pregnancy was based on a history of recent COVID-19 symptoms in addition to the following laboratorial tests: i) positive quantitative reverse transcription polymerase chain reaction (qRT–PCR) results for nasopharyngeal swab samples; ii) positive serum reactivity of IgM and/or IgG for SARS-CoV-2 (serological or rapid testing, Biomanguinhos, FIOCRUZ, Brazil) or iii) positive chest computed tomography (CT) results indicating pulmonary involvement suggestive of COVID-19. These criteria were recommended by the Ministry of Health of Brazil at the beginning of the COVID-19 pandemic [11].

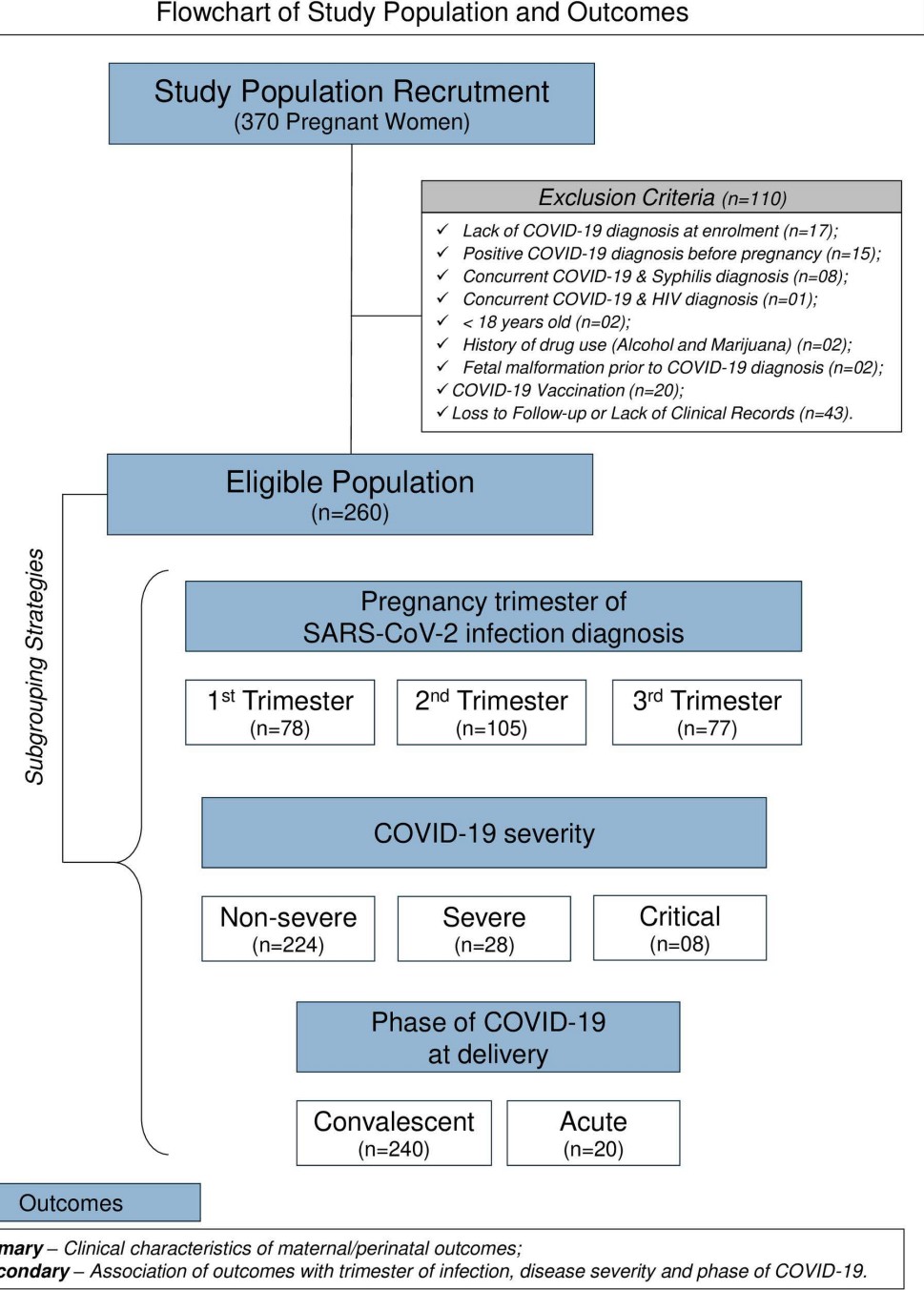

**Fig 1. Flowchart of study population and outcomes.** This is a prospective cohort study including pregnant women infected by SARS-CoV-2 during three pregnancy trimesters, referred as to *PR*egnancy *OU*tcome and child *D*evelopment – *E*ffects of *S*ARS-CoV-2 infection *T*rial (PROUDEST). Information about the study population recruitment and exclusion criteria is presented in the Material and Methods section. The protocol was designed to provide primary and secondary outcomes. Details regarding the deliverables included in the primary and secondary goals are presented in the main manuscript.

## Demographic and clinical records

Demographic and clinical records from participants, including epidemiological data, COVID-19 symptoms, COVID-19 severity, alcohol/illicit drug or cigarette use, a history of pregnancy diseases, actual pregnancy diseases or infections other than COVID-19, and other data were obtained at the first medical appointment by interview and questionnaires, followed by clinical and laboratorial tests to confirm the COVID-19 diagnosis.

Clinical obstetric data were recorded during prenatal follow-up, which occurred monthly up to 33 weeks of pregnancy, biweekly between 34–36 weeks of pregnancy, and weekly between 37 weeks of pregnancy and delivery. Data regarding childbirth was acquired during hospitalization and collected from medical records. Outpatient postpartum care was performed from 10 to 28 days after delivery. Intrauterine fetal evaluation was performed at least three times via serial ultrasounds: ≤ 14 weeks of pregnancy, between 20–24 weeks and 34–40 weeks of pregnancy. Efforts to minimize response bias were employed, consisting of medical appointments performed regularly in a short time interval.

As in any cohort study, follow-up losses were observed. To minimize follow-up losses, the participants were regularly contacted to continue in the study, proposing predelivery medical care, assistance at delivery and post-delivery follow-up, as well as the monitoring of newborns and children.

To preserve the anonymity of the participants, alphanumeric codes were used as identifiers in databases to enter, store and retrieve information. Ethnicity was classified as white, brown, black and indigenous, as self-reported by the participant. Pregnancy trimester was classified as: 1st = first trimester, comprising 4–13 weeks and 6 days of gestation; 2nd = second trimester, comprising 14 weeks to 27 weeks and 6 days of gestation and 3rd = third trimester, comprising 28–41 weeks and 6 days of gestation. The acute phase of COVID-19 was defined as the first 14 days of symptoms of SARS-CoV-2 infection at delivery. Convalescent phase included SARS-CoV-2 infection diagnosed at 1st, 2nd and 3rd pregnancy trimester.

The severity of the COVID-19 disease was classified as proposed by the WHO, referred as: non-severe, severe or critical [12]. Briefly, severe COVID-19 was defined as an oxygen saturation <90% in room air, a respiratory rate >30 breaths/min, > 30 breaths/min, or signs of severe respiratory distress (accessory muscle use, inability to complete full sentences), that were not life threatening. Critical COVID-19 was defined as acute respiratory distress syndrome (ARDS), sepsis, septic shock, or other conditions that normally require the provision of life-sustaining therapies, such as mechanical ventilation (invasive or non-invasive) or vasopressor therapy. Non-severe COVID-19 consisted of the absence of the aforementioned criteria for severe or critical COVID-19.

Pulmonary involvement on lung CT scans was defined as mild (<25% lung involvement), moderate (25–50% lung involvement) or severe (>50% lung involvement). Gestational diabetes mellitus (GDM) was defined as glucose intolerance first identified during pregnancy, with a fasting blood glucose level ≥ 92 mg/dL in any pregnancy trimester or a blood glucose level ≥ 180 mg/dL or ≥ 153 mg/dL 1 and 2 hours, respectively, after the 75 g oral glucose tolerance test (OGTT) between the 24th and 28th week of gestation, as proposed by the International Association of Diabetes and Pregnancy Study Groups Consensus Panel (IADPSG) [13]. Systemic arterial hypertension (SAH) was defined as a blood pressure greater than 140 x 90 mmHg during pregnancy measured on two separate occasions. Obesity was defined as body mass index ≥30 kg/m², and intrauterine fetal growth restriction was defined as an estimated fetal weight or abdominal circumference on ultrasound below the 10th percentile for gestational age.

Premature birth was defined as birth at <37 weeks of gestational age, full-term delivery was defined as delivery between 37−40 weeks and 6 days of gestational age, and post-term delivery was defined as delivery at ≥41 weeks of gestational age. Newborns were classified as adequate for gestational age (AGA), small for gestational age (SGA) or large for gestational age (LGA), according to the classification proposed by the International Fetal and Newborn Growth Consortium for the 21st Century (INTERGROWTH-21st) [14].

## Statistical analysis

Qualitative variables are presented by absolute (n) and relative (%) frequencies, and quantitative variables are presented as descriptive means and standard deviations.

The analyses were carried out in two stages: initially, bivariate analyses were performed to verify the isolated effect of each predictor on the outcomes, and subsequently, multivariate analyses adjusted by the pregestational control variables were performed to verify the effect of each predictor on the outcomes. The $p < 0.05$ was considered significant. The analyses were performed via SAS 9.4 software.

The Student's t-test or ANOVA was used to compare mean age values. Associations between qualitative variables were verified via Pearson's chi-square test, the Cochran-Armitage trend test or Fisher's exact test [15].

A generalized regression model was used to determine whether the severity of COVID-19 is a significant predictor of the occurrence of preterm or post-term birth in relation to the occurrence of full-term birth, adjusting for the effects of the pregestational control variables (GDM, hypertension, preeclampsia, asthma, obesity, education level, and maternal age).

Multivariate Poisson regression models with robust variance were employed to determine whether COVID-19 severity is a significant predictor for the occurrence of fetal distress and whether acute labor due to COVID-19 is a significant predictor for 1st minute scores ≤ 7. In both analyses, the effects of pregestational variables (GDM, hypertension, preeclampsia, asthma, obesity, education level and maternal age) were treated as control variables.

## Results

### Description of the study population

After applying the exclusion criteria, a total of 260 pregnant women infected with SARS-CoV-2 at distinct pregnancy trimester were selected for follow-up towards the postpartum period (Fig 1).

The mean age of the study population was $31.3 \pm 6.0$ years old (ranging from 18 to 52 years). The mean gestational age at COVID-19 diagnosis was $20.7 \pm 10.2$ weeks (ranging from 4 to 40 weeks). The ethnicity of participants was predominantly brown (60.8%). Education level comprised 11.9% of elementary school, 49.6% of high school and 38.5% of college degrees. Parity analysis demonstrated that most participants presented 2 or ≥3 pregnancies, including the pregnancy studied. The majority of COVID-19 diagnosis was performed by RT-PCR (60.4%). The study population was categorized according to the pregnancy trimester of SARS-CoV-2 infection diagnosis, comprising: 78 (30.0%) infected in the 1st trimester, 105 (40.4%) in the 2nd trimester and 77 in the 3rd trimester (29.6%) (Table 1).

### Clinical findings

Based on the clinical features, pregnant women were classified according to disease severity as follows: Non-Severe (86.2%), Severe (10.7%) and Critical (3.1%). Moreover, pregnant women were sub-grouped considering the phase of COVID-19 at delivery (Convalescent = 92.3% and Acute = 7.7%) (Table 2).

Of the 260 study participants, 42 patients (16.2%) required hospitalization due to COVID-19 (non-obstetric causes), 15 (5.8%) required oxygen supplementation, and 3 (1.1%) needed mechanical ventilation. Cesarean section was the most common delivery method (n = 158, 60.8%). A total of 40 patients experienced labor complications (6 of whom had more than one complication). Labor complications included: arterial hypertension (3.0%), intra- and postpartum hemorrhage (6.9%), oligohydramnios (2.3%), shoulder dystocia and polyhydramnios (0.8%), postpartum endometritis (0.8%), placental abruption (0.4%) and surgical wound infection (0.4%) (Table 2).

Maternal mortality comprised two participants who progressed to puerperal death (0.8%), including a 31-years-old patient who died due to respiratory failure associated with acute renal failure on the 18th postpartum day a 24-years-old woman who died 12 days postpartum due to tonic-clonic seizures, diffuse neurological impairment, and respiratory failure. Among the study population, one pregnant women experienced miscarriage that occurred spontaneously at 6 weeks and 5 days of pregnancy, with COVID-19 diagnosed at week 5 (Table 2).

**Table 1. Demographic and laboratorial features of the study population.**

| Parameter | Frequency n (%) |
|---|---|
| **Age** (years, mean ± SD) | 31.3 ± 6.0 |
| **Ethnicity** | |
| White | 73 (28.0) |
| Brown | 158 (60.8) |
| Black | 28 (10.8) |
| Indigenous | 1 (0.4) |
| **Education level** | |
| Elementary school | 31 (11.9) |
| High school | 129 (49.6) |
| College | 100 (38.5) |
| **Parity** | |
| 1 pregnancy | 78 (30.0) |
| 2 pregnancies | 74 (28.5) |
| ≥ 3 pregnancies | 108 (41.5) |
| **COVID-19 diagnosis** | |
| Rapid blood antibody test | 75 (28.8) |
| Serology | 25 (9.6) |
| RT–PCR | 157 (60.4) |
| Chest CT + clinical symptoms | 3 (1.2) |
| **Trimester of SARS-CoV-2 diagnosis*** | |
| 1st | 78 (30.0) |
| 2nd | 105 (40.4) |
| 3rd | 77 (29.6) |

RT–PCR = reverse transcription polymerase chain reaction; chest CT = chest computed tomography.

*1st = SARS-CoV-2 infection diagnosis at first pregnancy trimester (4–13 weeks and 6 days of gestation); 2nd = SARS-CoV-2 infection diagnosis at second pregnancy trimester (14 weeks to 27 weeks and 6 days of gestation); 3rd = SARS-CoV-2 infection diagnosis at third pregnancy trimester (28–41 weeks and 6 days of gestation). The results are expressed as the mean ± standard deviation (SD) or frequency (%).

## Current and previous gestational diseases

The analysis of adverse clinical conditions during current pregnancy demonstrated a total of 167 cases (64.2%), including: SAH (18.1%), gestational diabetes mellitus (35.8%), preeclampsia (6.2%) and intrauterine fetal growth restriction (10.4%). Of note was that patients with GDM comprised 30% of pregnant women with gestational diabetes diagnosed exclusively during the current pregnancy while 5.8% reported previous history of GDM (Table 2).

During prenatal follow-up, the participants were asked about their previous history of gestational diseases. Data demonstrated a total of 57 cases (21.9%), which included: SAH (6.2%) GDM (5.8%), preeclampsia (1.2%), pulmonary disease (5.8%), heart disease (2.3%) and intrauterine fetal growth restriction (0.8%) (Table 2).

## COVID-19 symptoms during pregnancy

The analysis of COVID-19 symptoms was carried out according to the pregnancy trimester of SARS-CoV-2 infection diagnosis and COVID-19 severity. The data are presented in the S1 Table. Anosmia (n = 168, 64.6%), nasal congestion/discharge (n = 160, 61.5%), headache (n = 158, 60.7%), ageusia (n = 152, 58.4%) and myalgia (n = 152, 58.4%) were the most common symptoms, followed by cough (n = 122, 46.9%), fever (n = 116, 44.6%) and dyspnea (n = 95, 36.5%). The less

**Table 2. Clinical features of women infected with SARS-CoV-2 during pregnancy.**

| Parameter | Frequency n (%) |
|---|---|
| **COVID-19 severity*** | |
| Non-Severe | 224 (86.2) |
| Severe | 28 (10.7) |
| Critical | 8 (3.1) |
| **Phase of COVID-19 at delivery**** | |
| Convalescent | 240 (92.3) |
| Acute | 20 (7.7) |
| **Chest tomography**** | |
| Convalescent | 240 (92.3) |
| Acute | 20 (7.7) |
| **Non-obstetric hospitalization** | 42 (16.2) |
| **Oxygen supplementation** | 15 (5.8) |
| **Mechanical ventilation** | 3 (1.1) |
| **Delivery** | |
| Vaginal | 102 (39.2) |
| Cesarean | 158 (60.8) |
| **Labor complications** | 40 (1.5) |
| **Maternal mortality** | 2 (0.8) |
| **Gestational diseases**** | |
| Current | 167 (64.2) |
| Previous | 57 (21.9) |

*COVID-19 severity was considered according to the World Health Organization classification [12].

**Acute phase = up to 14 days of symptoms onset at delivery; Convalescent phase = SARS-CoV-2 infection diagnosis at 1st, 2nd and 3rd pregnancy trimester.

***Gestational diseases included: systemic arterial hypertension, gestational diabetes mellitus, preeclampsia, pulmonary disease, heart disease and intrauterine fetal growth restriction.

common symptoms reported in the pregnant women studied were skin diseases (n = 9, 3.5%), joint pain (n = 11, 4.2%) and dizziness (n = 15, 5.8%), followed by nausea and vomiting (n = 17, 6.5%), diarrhea (n = 57, 21.9%), asthenia (n = 68, 26.1%) and sore throat (n = 74, 28.5%). Other symptoms (11, 4.2%) included periorbital pain, lumbar pain, hyporexia, abdominal pain, epigastric pain, otalgia, and chest pain (S1 Table).

The analysis of symptoms according to the pregnancy trimester of SARS-CoV-2 infection diagnosis demonstrated that women infected in the 2nd trimester experienced more myalgia ($p < 0.0001$), asthenia ($p = 0.0003$), ageusia ($p = 0.0117$), anosmia ($p < 0.0001$), headache ($p < 0.0001$) and joint pain ($p = 0.0015$) than those infected in the 1st and 3rd trimesters. Moreover, women infected in the 1st and 2nd trimesters presented more episodes of nasal congestion/discharge ($p = 0.0025$) and nausea ($p < 0.0001$) than those infected in the 3rd trimester. The other symptoms were not significantly associated with pregnancy trimester and SARS-CoV-2 infection (S1 Table).

In addition, pregnant women with critical COVID-19 had fever ($p = 0.0046$) as compared to those with non-severe or severe COVID-19. Dyspnea ($p < 0.0001$) was more common in pregnant women with severe or critical COVID-19 phases than in those with non-severe. Ageusia and headache were significantly more common in patients with non-severe disease than in patients with severe or critical disease ($p = 0.0303$ and $p = 0.0053$, respectively). The other symptoms showed no significant association with COVID-19 severity (S1 Table).

## Maternal outcomes according to the pregnancy trimester of SARS-CoV-2 infection, COVID-19 severity and phase of COVID-19 at delivery

The demographic/clinical parameters (age, ethnicity, education level and obesity) as well as the maternal outcomes (respiratory diseases, SAH, GDM, preeclampsia, gestational age at delivery and delivery routes) were evaluated according to the pregnancy trimester of SARS-CoV-2 infection, COVID-19 severity and phase of COVID-19 at delivery and the results presented in the Table 3.

No significant differences were observed for demographic/clinical parameters and maternal outcomes analyzed according to the pregnancy trimester of SARS-CoV-2 infection diagnosis. However, additional analysis demonstrated that while no women infected in the 1st trimester presented the critical form of COVID-19, three of them showed severe COVID-19 (3.9%) and the majority (96.1%) exhibited non-severe clinical form. Conversely, 7 (9.1%) and 15 (19.5%) pregnant women infected in the 3rd trimester had critical and severe forms of COVID-19, respectively (Table 4). Therefore, there is a significant association between the pregnancy trimester of SARS-CoV-2 infection occurs and the severity of COVID-19, as women infected in the 3rd trimester are more likely to have severe and critical disease, whereas those infected in the 1st or 2nd trimester are more likely to have non-severe disease (p < 0.0001) (Table 4).

The analysis of maternal outcomes according to COVID-19 severity demonstrated that while most women with non-severe COVID-19 experienced full-term birth, higher proportion of women with severe or critical COVID-19 faced pre-term delivery (p = 0.05) (Table 3).

The data regarding pregnancy outcomes according to phase of COVID-19 at delivery demonstrated that there was a significant association between the acute phase of COVID-19 at delivery and the occurrence of GDM (p = 0.04) and premature birth (p = 0.01) (Table 3). Additionally, the results demonstrated that pregnant women presenting acute phase of COVID-19 at delivery showed significantly higher disease severity (p < 0.0001) than those who were in the convalescent phase at delivery (Non-severe = 3.1% *vs.* 96.9%; Critical = 75.0% *vs.* 25.0%, respectively) (Table 5).

Multivariate analysis, without adjusting for the control variables, the phase of COVID-19 at delivery and COVID-19 severity were shown to be significant factors for the occurrence of preterm birth (OR = 3.14; 95% CI 1.18; 8.35; p = 0.0217) (Table 6). After adjusting the multivariate model according to age, education level, GDM status, arterial hypertension status, preeclampsia status, respiratory diseases status, and obesity status, there was an increase in the odds ratios, indicating that newborns to mothers with severe or critical COVID-19 at the time of delivery were 3.64 times more likely to be born preterm than those born to mothers without serious or critical COVID-19 (OR = 3.64; 95% CI 1.26; 10.53; p = 0.017) (Table 6).

## Newborn clinical findings

As part of this study, the newborns born to mother infected with SARS-CoV-2 at distinct pregnancy trimester were clinically monitored in the postpartum period (Fig 1).

Fetal growth restriction was observed in 0.8% of the cases. A total of 28 newborns were diagnosed with acute fetal distress (10.8%). The mean birth weight was 3,127 g ± 567 (ranging from 620 to 4,568 g). Most newborns (n = 187) were classified as adequate for gestational age (AGA = 71.9%), 36 were classified as small for gestational age (SGA = 13.8%) and 37 were classified as large for gestational age (LGA = 14.2%). The mean Apgar 1st minute score was 8 ± 1.2 (ranging from 2 to 10) and the 5th minute score was 9 ± 0.7 (ranging from 5 to 10). A total of 53 newborns (20.4%) has an Apgar 1st minute score ≤ 7 and 10 (3.8%) have an Apgar 5th minute score ≤ 7 (Table 7).

The newborn mortality rate includes two infants (0.8%), one stillbirth and a neonatal death (Table 7). First, the stillbirth occurred during week 31 of pregnancy in a 41-year-old woman with severe COVID-19 who was pregnant for the 4th time. The mother was diagnosed with chronic arterial hypertension and A1 GDM. The labor was induced at 38 weeks of gestation and the fetus was delivered with adequate weight for gestational age (AGA), albeit without any vital signs. Second,

**Table 3. Demographic and clinical features of SARS-CoV-2 infection according to pregnancy trimester of SARS-CoV-2 infection diagnosis, COVID-19 severity and phase of COVID-19 at delivery.**

| Parameters | Pregnancy trimester of SARS-CoV-2 infection diagnosis* n (%) | | | | COVID-19 severity** n (%) | | | | Phase of COVID-19 at delivery*** n (%) | | |
|---|---|---|---|---|---|---|---|---|---|---|---|
| | 1st n=78 (30.0%) | 2nd n=105 (40.4%) | 3rd n=77 (29.6%) | p | Non-severe n=224 (86.2%) | Severe n=28 (10.7%) | Critical n=8 (3.1%) | p | Conva-lescent n=240 (92.3%) | Acute n=20 (7.7%) | p |
| **Age** (years, mean±SD) | 30.7±6.1 | 32.1±5.8 | 30.8±6.0 | 0.22 | 31.2±5.8 | 31.8±7.4 | 31.4±6.1 | 0.89 | 31.3±6.0 | 30.4±6.1 | 0.51 |
| **Ethnicity** | | | | 0.23 | | | | 0.55 | | | 0.75 |
| White | 25 (32.1) | 32 (30.5) | 16 (20.8) | | 65 (29.0) | 7 (25.0) | 1 (12.5) | | 68 (28.3) | 5 (25.0) | |
| Brown | 43 (55.1) | 62 (59.0) | 53 (68.8) | | 134 (59.8) | 20 (71.4) | 5 (62.5) | | 144 (60) | 14 (70.0) | |
| Black | 10 (12.8) | 11 (10.5) | 7 (9.1) | | 24 (10.7) | 1 (3.6) | 2 (25.0) | | 27 (11.3) | 1 (5.0) | |
| Indigenous | 0 (0.0) | 0 (0.0) | 1 (1.3) | | 1 (0.5) | 0 (0.0) | 0 (0.0) | | 1 (0.4) | 0 (0.0) | |
| **Education level** | | | | 0.78 | | | | 0.92 | | | 0.15 |
| Elementary school | 11 (14.1) | 10 (9.5) | 10 (13.0) | | 26 (11.6) | 4 (14.3) | 1 (12.5) | | 29 (12.1) | 2 (10.0) | |
| High school | 35 (44.9) | 54 (51.4) | 40 (51.9) | | 110 (49.1) | 14 (50.0) | 5 (62.5) | | 115 (47.9) | 14 (70.0) | |
| College | 32 (41.0) | 41 (39.1) | 27 (35.1) | | 88 (39.3) | 10 (35.7) | 2 (25.0) | | 96 (40.0) | 4 (20.0) | |
| **Obesity** | | | | 0.78 | | | | 0.63 | | | 1.00 |
| No | 69 (88.5) | 96 (91.4) | 70 (90.9) | | 201 (89.7) | 26 (92.9) | 8 (100) | | 217 (90.4) | 18 (90.0) | |
| Yes | 9 (11.5) | 9 (8.6) | 7 (9.1) | | 23 (10.3) | 2 (7.1) | 0 (0.0) | | 23 (9.6) | 2 (10.0) | |
| **Respiratory diseases** | | | | 0.16 | | | | 0.23 | | | 0.39 |
| No | 71 (92.3) | 97 (92.4) | 76 (98.7) | | 209 (93.3) | 28 (100) | 8 (100) | | 225 (93.7) | 20 (100) | |
| Yes | 6 (7.7) | 8 (7.6) | 1 (1.3) | | 15 (6.7) | 0 (0.0) | 0 (0.0) | | 15 (6.2) | 0 (0.0) | |
| **SAH#** | | | | 0.42 | | | | 0.37 | | | 1.00 |
| No | 62 (79.5) | 90 (85.7) | 61 (79.2) | | 183 (81.7) | 22 (78.6) | 8 (100) | | 197 (82.1) | 16 (80.0) | |
| Yes | 16 (20.5) | 15 (14.3) | 16 (20.8) | | 41 (18.3) | 6 (21.4) | 0 (0.0) | | 43 (17.9) | 4 (20.0) | |
| **GDM#** | | | | 0.77 | | | | 0.09 | | | 0.04 |
| No | 49 (62.8) | 66 (62.9) | 52 (67.5) | | 140 (62.5) | 19 (67.9) | 8 (100) | | 150 (62.5) | 17 (85.0) | |
| Yes | 29 (37.2) | 39 (37.1) | 25 (32.5) | | 84 (37.5) | 9 (32.1) | 0 (0.0) | | 90 (37.5) | 3 (15.0) | |
| **Preeclampsia** | | | | 0.47 | | | | 0.44 | | | 0.62 |
| No | 72 (92.3) | 101 (96.2) | 71 (92.2) | | 211 (94.2) | 25 (89.3) | 8 (100) | | 226 (94.2) | 18 (90.0) | |
| Yes | 6 (7.7) | 4 (3.8) | 6 (7.8) | | 13 (5.8) | 3 (10.7) | 0 (0.0) | | 14 (5.8) | 2 (10.0) | |
| **Gestational age at delivery** | | | | 0.54 | | | | 0.05 | | | 0.01 |
| Pre-term | 9 (11.5) | 9 (8.6) | 12 (15.6) | | 21 (9.4) | 6 (21.4) | 3 (37.5) | | 23 (9.6) | 7 (35.0) | |
| Term | 66 (84.6) | 89 (84.8) | 60 (77.9) | | 189 (84.4) | 21 (75.0) | 5 (62.5) | | 202 (84.2) | 13 (65.0) | |
| Post-term | 3 (3.9) | 7 (6.6) | 5 (6.5) | | 14 (6.2) | 1 (3.6) | 0 (0.0) | | 15 (6.2) | 0 (0.0) | |
| **Delivery** | | | | 0.78 | | | | 0.37 | | | 0.38 |
| Vaginal | 33 (42.3) | 39 (37.1) | 30 (39.0) | | 86 (38.4) | 14 (50.0) | 2 (25.0) | | 96 (40.0) | 6 (30.0) | |
| Cesarean | 45 (57.7) | 66 (62.9) | 47 (61.0) | | 138 (61.6) | 14 (50.0) | 6 (75.0) | | 144 (60.0) | 14 (70.0) | |

#GDM=Gestational diabetes mellitus; SAH=Systemic arterial hypertension;

*1st=SARS-CoV-2 infection diagnosis at first pregnancy trimester (4–13 weeks and 6 days of gestation); 2nd=SARS-CoV-2 infection diagnosis at second pregnancy trimester (14 weeks to 27 weeks and 6 days of gestation); 3rd=SARS-CoV-2 infection diagnosis at third pregnancy trimester (28–41 weeks and 6 days of gestation).

**COVID-19 severity was considered according to the World Health Organization classification [12].

***Acute phase = up to 14 days of symptoms onset at delivery; Convalescent phase=SARS-CoV-2 infection diagnosis at 1st, 2nd and 3rd pregnancy trimester. The results are expressed as the mean±standard deviation (SD) or frequency (%). Comparative analyses were carried out by ANOVA or Chi-square/Fisher tests. Significance was considered at p-values<0.05.

**Table 4. COVID-19 severity according to the pregnancy trimester of SARS-CoV-2 infection diagnosis.**

| COVID-19 severity | Total n (%) | Pregnancy trimester of SARS-CoV-2 infection diagnosis* | | | |
| --- | --- | --- | --- | --- | --- |
| | | 1st n (%) | 2nd n (%) | 3rd n (%) | p |
| Non-severe | 224 (86.2) | 75 (96.1) | 94 (89.5) | 55 (71.4) | < 0.0001 |
| Severe | 28 (10.7) | 3 (3.9) | 10 (9.5) | 15 (19.5) | |
| Critical | 8 (3.1) | 0 (0.0) | 1 (1.0) | 7 (9.1) | |

*1st = SARS-CoV-2 infection diagnosis at first pregnancy trimester (4–13 weeks and 6 days of gestation); 2nd = SARS-CoV-2 infection diagnosis at second pregnancy trimester (14 weeks to 27 weeks and 6 days of gestation); 3rd = SARS-CoV-2 infection diagnosis at third pregnancy trimester (28–41 weeks and 6 days of gestation). COVID-19 severity according to the World Health Organization classification [12]. Comparative analysis was carried out by Chi-square test. Significance was considered at p-values < 0.05.

**Table 5. COVID-19 severity according to the phase of COVID-19 at delivery.**

| COVID-19 severity | Total n (%) | Phase of COVID-19 at delivery* | | |
| --- | --- | --- | --- | --- |
| | | Convalescent n (%) | s n (%) | p |
| Non-severe | 224 (86.2) | 217 (96.9) | 7 (3.1) | < 0.0001 |
| Severe | 28 (10.7) | 21 (75.0) | 7 (25.0) | |
| Critical | 8 (3.1) | 2 (25.0) | 6 (75.0) | |

*Acute phase = up to 14 days of symptoms onset at delivery; Convalescent phase = SARS-CoV-2 infection diagnosis at 1st, 2nd and 3rd pregnancy trimester. COVID-19 severity according to the World Health Organization classification [12]; Comparative analysis was carried out by Cochran–Armitage trend test. Significance was considered at p-values < 0.05.

**Table 6. Odds ratio modeling of gestational age at delivery according to COVID-19 severity.**

| Outcome | Comparisons | Factors | OR not adjusted* (95% CI) | p | OR adjusted** (95% CI) | p |
| --- | --- | --- | --- | --- | --- | --- |
| Gestational age at delivery | Preterm vs. Term | Severe & Critical vs. Non-severe | 3.14 (1.18 - 8.35) | 0.0217 | 3.64 (1.26 - 10.53) | 0.017 |

OR = Odds Ratio. COVID-19 severity was considered according to the World Health Organization classification [12].

*Generalized logit model.

**The 95% confidence intervals (95% CI) adjusted for age, education level, gestational diabetes mellitus, systemic arterial hypertension, pre-eclampsia, respiratory disease and obesity.

the neonatal death occurred at 11 days of life due to extreme prematurity (26 weeks and 1 day of gestation), respiratory failure, and neonatal sepsis. The baby was born to a 26-year-old primiparous woman diagnosed with mild COVID-19 and isthmus-cervical incompetence. The cause of the newborn death, determined after necropsy and placental analysis, was hypoxia and hemorrhage of multiple organs.

Fetal growth restriction, acute fetal distress, birth weight classification (SGA, AGA or LGA) and Apgar 1st minute score ≤ 7 did not differ according to pregnancy trimester of SARS-CoV-2 infection diagnosis (Table 8).

Data analysis demonstrated a significant association between the presence of acute fetal distress with disease severity (p = 0.01) (Table 8). Moreover, there was a significant relationship between COVID-19 severity and newborn weight classification (p = 0.01) with non-severe COVID-19 associated with AGA and critical disease associated with LGA (Table 8).

**Table 7. Clinical features of newborns born to mothers infected with SARS-CoV-2 during pregnancy.**

| Parameter | Frequency n (%) |
|---|---|
| **Fetal growth restriction** | |
| No | 258 (99.2) |
| Yes | 2 (0.8) |
| **Acute fetal distress** | |
| No | 232 (89.2) |
| Yes | 28 (10.8) |
| **Birth weight#** | |
| Weight (g, mean ± SD) | 3,127 ± 567 |
| Small (SGA) | 36 (13.8) |
| Adequate (AGA) | 187 (71.9) |
| Large (LGA) | 37 (14.2) |
| **Apgar 1st minute score** | |
| Score (mean ± SD) | 8 ± 1.2 |
| ≤ 7 | 53 (20.4) |
| > 7 | 207 (79.6) |
| **Apgar 5th minute score** | |
| Score (mean ± SD) | 9 ± 0.7 |
| ≤ 7 | 10 (3.8) |
| > 7 | 250 (96.2) |
| **Newborn mortality** | 2 (0.8) |

#Birth Weight were classified as small (SGA), adequate (AGA) and large (LGA) for gestational age. The results are expressed as the mean ± standard deviation (SD) or frequency (%). The newborn mortality includes stillbirth and neonatal death.

Acute fetal distress was more common (p = 0.04) in pregnant women who were in the acute phase of COVID-19 (25.0%) as compared to women in the convalescent phase (9.6%) (Table 8). These findings indicate that pregnant women with severe and critical disease and those with acute COVID-19 were more likely to have newborns diagnosed with fetal distress at the time of delivery.

Modeling counts of acute fetal distress adjusted by age, education level, GDM status, arterial hypertension status, preeclampsia status, respiratory disease status and obesity status, showed that newborns born to mothers with severe or critical COVID-19 were more prone to acute fetal distress, with a 2.4 times higher risk than those born to mothers with non-severe COVID-19 (p = 0.036) (Table 9).

Regardless of no differences were observed for Apgar 1st minute score according to pregnancy trimester of SARS-CoV-2 infection diagnosis, COVID-19 severity and phase of COVID-19 at delivery (Table 8), the prevalence ratio analysis demonstrated an association between the phase of COVID-19 with the Apgar 1st minute score ≤ 7 (Table 9). Modeling counts adjusted for age, education level, GDM status, arterial hypertension status, preeclampsia status, respiratory disease status, and obesity status, showed that newborns born to mothers with acute phase of COVID-19 at delivery were 2.56 times more likely to have an Apgar 1st minute score ≤ 7 than newborns born to mothers with convalescent phase of COVID-19 at delivery (p = 0.018) (Table 9).

## Discussion

Different symptoms of COVID-19 were observed in the patients studied, varying according to the trimester of SARS-CoV-2 infection occurred and disease severity. Anosmia, runny nose and/or nasal congestion, headache, ageusia and myalgia

Table 8. Clinical features of newborns born to mothers with SARS-CoV-2 infection according to pregnancy trimester of SARS-CoV-2 infection diagnosis, COVID-19 severity and phase of COVID-19 at delivery.

| Parameters | Pregnancy trimester of SARS-CoV-2 infection diagnosis* n (%) | | | | COVID-19 severity** n (%) | | | | Phase of COVID-19 at delivery*** n (%) | | |
|---|---|---|---|---|---|---|---|---|---|---|---|
| | 1st n=78 (30.0%) | 2nd n=105 (40.4%) | 3rd n=77 (29.6%) | p | Non-severe n=224 (86.2%) | Severe n=28 (10.7%) | Critical n=8 (3.1%) | p | Convalescent n=240 (92.3%) | Acute n=20 (7.7%) | p |
| Fetal growth restriction | | | | 0.52 | | | | 1.00 | | | 0.15 |
| No | 77 (98.7) | 105 (100) | 76 (98.7) | | 222 (99.1) | 28 (100) | 8 (100) | | 239 (99.6) | 19 (95.0) | |
| Yes | 1 (1.2) | 0 (0.0) | 1 (1.3) | | 2 (0.9) | 0 (0.0) | 0 (0.0) | | 1 (0.4) | 1 (5.0) | |
| Acute fetal distress | | | | 0.41 | | | | 0.01 | | | 0.04 |
| No | 72 (92.3) | 94 (89.5) | 66 (85.7) | | 204 (91.1) | 23 (82.1) | 5 (62.5) | | 217 (90.4) | 15 (75.0) | |
| Yes | 6 (7.7) | 11 (10.5) | 11 (14.3) | | 20 (8.9) | 5 (17.9) | 3 (37.5) | | 23 (9.6) | 5 (25.0) | |
| Birth weight # | | | | 0.86 | | | | 0.01 | | | 0.29 |
| Small (SGA) | 12 (15.4) | 14 (13.3) | 10 (13.0) | | 28 (12.5) | 7 (25) | 1 (12.5) | | 32 (13.3) | 4 (20.0) | |
| Adequate (AGA) | 53 (67.9) | 78 (74.3) | 56 (72.7) | | 166 (74.1) | 18 (64.3) | 3 (37.5) | | 176 (73.4) | 11 (55.0) | |
| Large (LGA) | 13 (16.7) | 13 (12.4) | 11 (14.3) | | 30 (13.4) | 3 (10.7) | 4 (50.0) | | 32 (13.3) | 5 (25.0) | |
| Apgar 1st minute score | | | | 0.54 | | | | 0.32 | | | 0.25 |
| ≤7 | 19 (24.4) | 19 (18.1) | 15 (19.5) | | 42 (18.8) | 9 (32.1) | 2 (25.0) | | 47 (19.6) | 6 (30.0) | |
| >7 | 59 (75.6) | 86 (81.9) | 62 (80.5) | | 182 (81.2) | 19 (67.9) | 6 (75.0) | | 193 (80.4) | 14 (70.0) | |

#Birth Weight were classified as small (SGA), adequate (AGA) and large (LGA) for gestational age.

*1st = SARS-CoV-2 infection diagnosis at first pregnancy trimester (4–13 weeks and 6 days of gestation); 2nd = SARS-CoV-2 infection diagnosis at second pregnancy trimester (14 weeks to 27 weeks and 6 days of gestation); 3rd = SARS-CoV-2 infection diagnosis at third pregnancy trimester (28–41 weeks and 6 days of gestation).

**COVID-19 severity was considered according to the World Health Organization classification [12].

***Acute phase = up to 14 days of symptoms onset at delivery; Convalescent phase = SARS-CoV-2 infection diagnosis at 1st, 2nd and 3rd pregnancy trimester. The results are expressed as the mean ± standard deviation (SD) or frequency (%). Comparative analyses were carried out by ANOVA or Chi-square/Fisher tests. Significance was considered at p-values < 0.05.

Table 9. Modeling counts of acute fetal distress and Apgar 1st minute score ≤7 according to the COVID-19 severity and phase of infection.

| Outcome | Factors | PR not adjusted* (95% CI) | p | PR adjusted** (95% CI) | p |
|---|---|---|---|---|---|
| Acute fetal distress | Severe & Critical vs. Non-severe | 2.87 (1.32 - 6.21) | 0.007 | 2.40 (1.06 - 5.46) | 0.036 |
| Apgar 1st minute score ≤7 | Acute phase vs. Convalescent phase | 2.32 (1.07 - 5.05) | 0.032 | 2.56 (1.17 - 5.56) | 0.018 |

PR = Prevalence Ratio. COVID-19 severity was considered according to the World Health Organization classification [12]. Acute phase = up to 14 days of symptoms onset at delivery; Convalescent phase = SARS-CoV-2 infection diagnosis at 1st, 2nd and 3rd pregnancy trimester.

*Poisson regression model.

**Prevalence ratio adjusted by setting the 95% confidence interval (95% CI), adjusted for age, education level, gestational diabetes mellitus, systemic arterial hypertension, pre-eclampsia, respiratory disease and obesity.

were the most common symptoms reported during the acute phase of COVID-19 and, in general, were associated with non-severe disease. Moreover, fever and dyspnea were not common but were associated with disease severity. In the PregCOV-19 systematic review, cough and fever were the most common COVID-19 symptoms in pregnant women,

whereas dyspnea, myalgia, ageusia, and diarrhea were less common [16]. Additionally, the PRIORITY study revealed that cough, sore throat, myalgia and fever were the most prevalent initial symptoms in infected women [17].

Since pregnancy causes mechanical, physiological, and immunological changes in women's bodies, pregnancy potentially increases susceptibility to infectious diseases, associated complications, and adverse outcomes, including an altered response to COVID-19 infection [18–20]. Reports of maternal mortality related to COVID-19 in low- and middle-income countries exceeded global figures; possible reasons include the aggravation of SARS-CoV-2 infection related to the changes caused by pregnancy or due to interruptions in access to maternity services leading to delayed healthcare. Brazil, especially, was hit hard by the COVID-19 pandemic, with a high number of daily reported cases and maternal deaths, especially in postpartum women [21–26]. The mortality rate among pregnant Brazilian women was approximately 5% in 2020 and 10% in 2021, showing a progressive decrease after the vaccination campaign started in April 2021 [4].

In the present study, which included only pregnant women who had not been vaccinated, the mortality rate was 0.8% (2 out of 260 women); deaths exclusively occurred during puerperium. If we consider only participants with severe and critical disease (36 of the 260 pregnant women), the mortality rate was 5.6% (2 of 36 women). A retrospective cohort study including 1,386 pregnant Brazilian women with a confirmed diagnosis of COVID-19 also revealed that women infected with COVID-19 during pregnancy had a greater risk of maternal death (relative risk = 18.7; 95% CI = 11.1–31.7) [21].

According to data from the European Centre for Disease Prevention and Control (ECDC) and the GISAID platform, the Alpha variant predominated from December 2020 to April 2021, being succeeded by Delta between June and December 2021, and later by Omicron, which became dominant from January 2022 [27,28]. Specifically, our study recruited patients between May 2020 and May 2021 – a period during which multiple SARS-CoV-2 lineages were circulating in Brazil. Early in this period, B.1.1.28 and B.1.1.33 lineages predominated; by late 2020, the VOI Zeta (P.2) emerged, followed by the VOC Gamma (P.1), first detected in Manaus, Brazil in November 2020 and becoming the dominant variant during the severe second wave in early 2021. In May 2021, the VOC Alpha (B.1.1.7) was also present, although with more limited circulation. These variants were characterized by distinct mutations, particularly in the spike protein, associated with increased transmissibility, immune escape, and, in some cases, higher clinical severity [29–31].

Variant epidemiology analyses highlight how genomic surveillance has been instrumental in tracking variant emergence, understanding transmission patterns, and informing public health interventions. For clinicians, particularly those caring for pregnant women in their third trimester, awareness of the predominant circulating variants is critical, as certain VOCs have been linked to increased risk of reinfection, and more severe disease presentations [29–31]. There is evidence that both maternal and neonatal outcomes were worse during the Delta wave of the SARS-CoV-2 pandemic than in preceding periods, but our study included mostly pregnant women with the Alpha or pre-Delta variant [32]. Regarding disease severity, more critical and severe cases of COVID-19 were detected in women who were infected in the 3rd trimester of pregnancy or in the acute phase of COVID-19 during delivery. Previous studies have demonstrated that COVID-19 disease severity is higher in late pregnancy as compared to early pregnancy. However, the complete immunopathological mechanisms of this condition are not completely defined [19,33].

According to our findings, SARS-CoV-2 infection during pregnancy is associated with adverse maternal and neonatal outcomes, especially among pregnant women infected by SARS-CoV-2 in the 3rd trimester, severe and critical disease and acute phase of SARS-CoV-2 infection at delivery, ultimately linked with preterm labor, cesarean delivery, acute fetal distress and Apgar 1st minute score ≤ 7.

Previous studies also suggested that SARS-CoV-2 infection during pregnancy is associated with several adverse pregnancy outcomes including preeclampsia, preterm birth, cesarean delivery, stillbirth and small birth weight (SGA), especially among pregnant women with severe COVID-19 or with COVID-19 at the time of delivery [34–39]. The INTERCOVID Multinational Cohort Study, comprising 43 centers in 18 countries, revealed that COVID-19 was associated with increased risks for preeclampsia/eclampsia (1.7-fold), preterm birth (1.6-fold), small birth weight (1.5-fold) and severe neonatal morbidity (2.6-fold) [9]. The COVI-PREG registry revealed increased risks of adverse and maternal outcomes, including

oxygen requirements, hospitalization, intensive care unit (ICU) admission and premature delivery in pregnant women infected with SARS-CoV-2 [8].

Premature labor, acute fetal distress and Apgar 1$^{st}$ minute score ≤ 7 were associated with the acute phase of COVID-19 at delivery. COVID-19 severity may be explained by multiple mechanisms underlying the adverse effects of SARS-CoV-2 infection on the placenta and developing fetus, including hypoxia-related issues, an excessive immune response, altered inflammation and thrombosis.

Placental damage, evidenced by histological abnormalities linked to SARS-CoV-2 transplacental transmission, could compromise fetal blood flow and lead to placental insufficiency. These abnormalities, primarily characterized by significant perivillous fibrin deposition and chronic histiocytic intervillositis, are referred to as SARS-CoV-2 placentitis and can occur even without placental infection [40,41]. Changes at the maternal–fetal interface during systemic infection linked to an inflammatory condition in the placenta may serve to protect the placenta and fetus from infection; however, these changes also have the potential to drive pathological changes with adverse consequences for offspring, such as hypoxia and fetal distress [42].

Another explanation is that SARS-CoV-2 infection during pregnancy can induce changes in the maternal immune response, with effects on pregnancy outcomes. A paper published by our group demonstrated that, in general, serum soluble mediators have different trajectories during a healthy pregnancy and are disturbed in pregnant women with convalescent SARS-CoV-2 infection. Higher levels of most soluble mediators were observed in the COVID-19 group than in the healthy control group, with major increases in the levels of proinflammatory cytokines, including IL-6, TNF-α and IFN-γ. These findings may also contribute to explaining the clinical course of pregnancy complications [20].

One specific adverse maternal outcome observed in pregnant women affected by COVID-19 in this cohort study attracted our attention: gestational diabetes mellitus. Our data demonstrated that high number of pregnant women infected with SARS-CoV-2 presented with GDM (93 out of 260, 35.8%). In a meta-analysis of women before the COVID-19 pandemic that used the IADPSG's diagnostic criteria for GDM, the global prevalence of GDM was 14.2% (95% CI = 14.2–14.3%). In low-income countries, middle-income countries and high-income countries, the prevalence rates were 14.7% (95% CI = 12.9–16.7%), 9.9% (95% CI = 9.0–10.1%) and 14.4% (95% CI = 14.3–14.4%), respectively, and these rates were much lower than those observed in the present study [13]. Other studies also reported a significantly increased prevalence of GDM in pregnant women during the COVID-19 pandemic [36,43]. Previous studies suggested that SARS-CoV-2 may trigger hyperglycemia and diabetes mellitus through virus interaction with angiotensin-converting enzyme 2 (ACE2) receptors, resulting damage to the pancreatic islet cells [44]. This outcome can increase the risk of short-term adverse maternal and perinatal outcomes, including shoulder dystocia, fetal macrosomia, cesarean section, hypertensive disorders, neonatal hypoglycemia, and admission of neonates to the intensive care unit [44].

According to our results, the prevalence of preeclampsia in pregnant women infected with SARS-CoV-2 was similar to that previously reported in women not infected with SARS-CoV-2 [45,46]. Moreover, studies have demonstrated an increased risk of preeclampsia in pregnant women infected with SARS-CoV-2, suggesting that COVID-19, characterized by virus-induced effects on endothelial cells via the ACE2 receptor and TMPRSS2 coreceptor, can cause endothelial dysfunction and disrupt vascular integrity, leading to hyperinflammation and hypercoagulability, with immunopathological mechanisms overlapping between COVID-19 and preeclampsia [47].

Strategies to reduce the risk of poor obstetric outcomes related to COVID-19 should be developed. Multiple factors associated with poor obstetric outcomes must be considered, such as limited access to care, inadequate standardization of care, racial and ethnic discrimination, social and economic disparities, and misinformation regarding with vaccination [41,48,49].

## Strengths and limitations

The main strength of the present study was the design comprising a prospective cohort follow-up of a representative number of pregnant women at the beginning of the pandemic, during which data collection was scarce. Our study followed

a strict protocol, excluding other infectious diseases or vaccination against SARS-CoV-2 that could result in selection bias. Epidemiological and clinical data from the pregnancy, childbirth, and postpartum periods were collected regularly despite all the challenges associated with combating the COVID-19 pandemic.

The present study has some limitations. The sample did not include asymptomatic pregnant women because tests were performed exclusively on symptomatic patients at the beginning of the pandemic due to a lack of public resources. This study was conducted during the predominant circulation of the SARS-CoV-2 Alpha variant; therefore, the impact of other variants on obstetrics outcomes may be different from those reported in this study.

However, the overall interpretation of the results obtained in this study should be performed cautiously. Since the data obtained in this research may not be obtained in populations in other sociodemographic contexts, further studies should be carried out with the aim of evaluating the clinical and obstetric outcomes of women infected with SARS-CoV-2 during any trimester of pregnancy and with any SARS-CoV-2 strains and of establishing the pathophysiological mechanisms of COVID-19 severity during pregnancy.

Furthermore, unanswered questions about the direct viral effect and the effects of the placental inflammatory response and maternal cytokine storm caused by maternal SARS-CoV-2 infection on the fetus and newborns need to be evaluated through long-term studies. Understanding these phenomena should contribute to the proper management of newborns born to SARS-CoV-2-infected mothers.

## Conclusions

According to our study, SARS-CoV-2 infection during pregnancy is associated with numerous adverse maternal and neonatal outcomes, including preterm labor, acute fetal distress and a Apgar $1^{st}$ minute score $\leq 7$, especially among pregnant women infected in the 3rd trimester, those in the acute phase of COVID-19 at childbirth, and those with severe and critical COVID-19.

Considering that SARS-CoV-2 is constantly mutating, and several countries worldwide have not yet established complete vaccination for all pregnant women, there is a need to maintain data collection regarding the effects of SARS-CoV-2 infection during pregnancy in order to decrease maternal and neonatal risks. The identification of pregnant women as a particularly affected group can serve as an alert not only to monitor the evolution of SARS-CoV-2, but also to monitor new emerging pathogens.

Understanding the clinical symptoms, severity, and possible adverse obstetric outcomes of COVID-19 during pregnancy can help establish the most suitable approach for decreasing maternal and neonatal risk in this population.

## Supporting information

**S1 Table. COVID-19 symptoms according to the pregnancy trimester of SARS-CoV-2 infection diagnosis and COVID-19 severity.**
(DOCX)

## Acknowledgments

We express our gratitude to all the pregnant women and their families as well as all the students, residents, and healthcare professionals whose support was essential to conduct this study. We are also grateful to the Nucleus of Support on Research from the Sabin Institute for performing laboratory tests. This research was performed by students and professors enrolled in the Programa de Pós-graduação em Ciências Médicas da Universidade de Brasília (UnB), supported by the Coordenação de Aperfeiçoamento de Pessoal de Nível Superior (CAPES). JGACR, OTN and OAMF received PQ fellowships from CNPq. OAMF participated in the fellow program supported by the Universidade do Estado do Amazonas (PROVISIT N° 005/2023-PROPESP/UEA).

## Author contributions

**Conceptualization:** Lizandra Paravidine Sasaki, Geraldo Magela Fernandes, Maria Eduarda Canellas de Castro, Patricia Shu Kurizky, Cleandro Pires de Albuquerque, Alberto Moreno Zaconeta, Licia Maria Henrique da Mota.

**Data curation:** Lizandra Paravidine Sasaki, Geraldo Magela Fernandes, Ângelo Pereira da Silva, Yacara Ribeiro Pereira, Aleida Oliveira de Carvalho, Felipe Motta, David Alves de Araújo Junior, Gabriela Profírio Jardim-Santos, Heidi Luise Schulte, Clara Correia de Siracusa, Isadora Pastrana Rabelo, Pedro Sadi Monteiro.

**Formal analysis:** Lizandra Paravidine Sasaki, Ismael Artur Costa-Rocha, Jordana Grazziela Alves Coelho-dos-Reis, Olindo Assis Martins-Filho, Licia Maria Henrique da Mota.

**Funding acquisition:** Lizandra Paravidine Sasaki, Dayde Lane Mendonça Da Silva, Laila Salmen Espindola, Licia Maria Henrique da Mota.

**Investigation:** Lizandra Paravidine Sasaki, Geraldo Magela Fernandes, Ângelo Pereira da Silva, Yacara Ribeiro Pereira, Aleida Oliveira de Carvalho, Felipe Motta, David Alves de Araújo Junior, Maria Eduarda Canellas de Castro, Clara Correia de Siracusa, Isadora Pastrana Rabelo, Pedro Sadi Monteiro.

**Methodology:** Lizandra Paravidine Sasaki, Geraldo Magela Fernandes, Ângelo Pereira da Silva, Yacara Ribeiro Pereira, Aleida Oliveira de Carvalho, Felipe Motta, David Alves de Araújo Junior, Maria Eduarda Canellas de Castro, Cleandro Pires de Albuquerque, Alberto Moreno Zaconeta, Licia Maria Henrique da Mota.

**Project administration:** Lizandra Paravidine Sasaki, Geraldo Magela Fernandes, Cleandro Pires de Albuquerque, Alberto Moreno Zaconeta, Licia Maria Henrique da Mota.

**Resources:** Lizandra Paravidine Sasaki, Geraldo Magela Fernandes, Felipe Motta, David Alves de Araújo Junior, Maria Eduarda Canellas de Castro, Agenor de Castro Moreira dos Santos Junior.

**Supervision:** Cleandro Pires de Albuquerque, Laila Salmen Espindola, Olindo Assis Martins-Filho, Alberto Moreno Zaconeta, Licia Maria Henrique da Mota.

**Validation:** Geraldo Magela Fernandes, Fabiola Cristina Ribeiro Zucchi, Rosana Tristão, Dayde Lane Mendonça Da Silva, Otávio de Toledo Nóbrega, Alexandre Anderson de Sousa Munhoz Soares, Ciro Martins Gomes.

**Visualization:** Lizandra Paravidine Sasaki, Maria Eduarda Canellas de Castro, Ismael Artur Costa-Rocha, Jordana Grazziela Alves Coelho-dos-Reis, Cleandro Pires de Albuquerque, Olindo Assis Martins-Filho, Alberto Moreno Zaconeta, Licia Maria Henrique da Mota.

**Writing – original draft:** Lizandra Paravidine Sasaki, Geraldo Magela Fernandes, Ângelo Pereira da Silva, Yacara Ribeiro Pereira, Aleida Oliveira de Carvalho, Felipe Motta, David Alves de Araújo Junior, Maria Eduarda Canellas de Castro, Gabriela Profírio Jardim-Santos, Heidi Luise Schulte, Ismael Artur Costa-Rocha, Jordana Grazziela Alves Coelho-dos-Reis, Olindo Assis Martins-Filho.

**Writing – review & editing:** Lizandra Paravidine Sasaki, Geraldo Magela Fernandes, Maria Eduarda Canellas de Castro, Ismael Artur Costa-Rocha, Jordana Grazziela Alves Coelho-dos-Reis, Cleandro Pires de Albuquerque, Olindo Assis Martins-Filho, Alberto Moreno Zaconeta, Licia Maria Henrique da Mota.

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
