## [Decision Letter · Decision Letter 0]

PONE-D-25-15315Clinical characteristic and outcomes of pregnant women with COVID‐19: The PROUDEST prospective cohort studyPLOS ONE

Dear Dr. Sasaki,

Thank you for submitting your manuscript to PLOS ONE. After careful consideration, we feel that it has merit but does not fully meet PLOS ONE’s publication criteria as it currently stands. Therefore, we invite you to submit a revised version of the manuscript that addresses the points raised during the review process. The manuscript meets most of the PLOS ONE publication criteria, with well-described methods, appropriate statistical analyses, and clearly reported results. However, some items still require revision to fully meet the publication criteria, specifically:

**Criteria 3 - Experiments, statistics, and other analyses are performed to a high technical standard and are described in sufficient detail:** Although the overall study design is appropriate, the recruitment strategy—described as non-probabilistic convenience sampling—lacks operational detail. Additional clarification is required regarding how media and social networks were used to recruit participants. This information is important to assess potential selection bias and the generalizability of the findings.

**Criteria 4 - Conclusions are presented in an appropriate fashion and are supported by the data:** Given the long study period and the known circulation of multiple SARS-CoV-2 variants in Brazil, the discussion should include a brief contextualization of the genomic landscape (e.g., VOCs/VOIs). Including this point will strengthen the interpretation of findings and increase clinical relevance, particularly for pregnant populations. Reference to appropriate literature (e.g., Male 2022) is encouraged.

**Criteria 7 - The article adheres to appropriate reporting guidelines and community standards for data availability:** The manuscript does not include a formal Data Availability Statement, as required by PLOS ONE policy. This is essential for ensuring transparency and reproducibility.

We look forward to receiving your revised manuscript.

Kind regards,

Elma Izze Da Silva Magalhães

Academic Editor

PLOS ONE

Journal Requirements:

Reviewers' comments:

Reviewer's Responses to Questions

**Comments to the Author**

1. Is the manuscript technically sound, and do the data support the conclusions?

Reviewer #1: Yes

Reviewer #2: Yes

2. Has the statistical analysis been performed appropriately and rigorously? 

Reviewer #1: Yes

Reviewer #2: Yes

3. Have the authors made all data underlying the findings in their manuscript fully available?

Reviewer #1: No

Reviewer #2: Yes

4. Is the manuscript presented in an intelligible fashion and written in standard English?

Reviewer #1: Yes

Reviewer #2: Yes

5. Review Comments to the Author

Reviewer #1: The manuscript clearly outlines a prospective cohort study in the methods, detailing the ethical approvals, study setting, and inclusion/exclusion criteria.

However, the recruitment strategy, while described as non-probabilistic convenience sampling, could benefit from a slightly more detailed explanation of how media and social networks were utilized.

The diagnostic criteria for SARS-CoV-2 infection, incorporating RT-PCR, serology, and chest CT findings as per Brazilian Ministry of Health guidelines, appear appropriate for the study period.

The Demographical and clinical records subsection describes the data collection process, including interviews, questionnaires, and follow-up schedules across pregnancy trimesters. The efforts to minimize response bias and follow-up losses are noted. The definitions of key variables, such as ethnicity, pregnancy trimester, COVID-19 phase, and severity (based on WHO criteria), are explicitly provided. The definitions for obstetric complications like GDM (IADPSG criteria), SAH, obesity, and fetal growth restriction are also clearly stated and reference relevant guidelines where applicable. Similarly, the definitions for premature birth and newborn classifications (INTERGROWTH-21st) enhance the rigor of the study.

The Statistical analysis outlines the methods used, including bivariate and multivariate analyses. The adjustment for pregestational control variables demonstrates an attempt to account for potential confounding factors.

The Results section systematically presents the findings, referencing specific tables. The description of labor complications and maternal mortality provides critical outcome data. The analysis of current and previous gestational diseases offers valuable clinical history.

However, , the manuscript lacks a clear statement regarding the availability of the raw data underlying these findings. The PLOS ONE data policy requires this information to be explicitly stated, either with a link to a repository or a statement that the data are within the supplementary information. The absence of this (as the abstract and introduction do not mention it) is a significant concern that needs to be addressed to ensure the reproducibility and transparency of the research.

Reviewer #2: Manuscript is well written and is covering all aspects already. I just have an optional suggestion that if genomic surveillance of variants could briefly be discussed? such as VOC/VOI because a several variants of SARS-CoV-2 were circulating in Brazil given the time span of the study so if it briefly describes will help clinicians for targeted intervention especially for the women in their third trimester. The thing I’m suggesting has been described by (Male 2022) so adding this aspect with respect to current study will enhance the impact of this study further.

6. PLOS authors have the option to publish the peer review history of their article (what does this mean? ). If published, this will include your full peer review and any attached files.

**Do you want your identity to be public for this peer review?** For information about this choice, including consent withdrawal, please see our Privacy Policy .

Reviewer #1: No

Reviewer #2: **Yes: ** Bhatti, T. H.

---

## [Author Response · Author response to Decision Letter 1]

6 Jun 2025

Editor Comments:

The manuscript meets most of the PLOS ONE publication criteria, with well-described methods, appropriate statistical analyses, and clearly reported results. However, some items still require revision to fully meet the publication criteria, specifically:

Criteria 3 - Experiments, statistics, and other analyses are performed to a high technical standard and are described in sufficient detail: Although the overall study design is appropriate, the recruitment strategy - described as non-probabilistic convenience sampling - lacks operational detail. Additional clarification is required regarding how media and social networks were used to recruit participants. This information is important to assess potential selection bias and the generalizability of the findings.

Dear Editor,

Thank you for your thoughtful comment regarding the recruitment strategy. We appreciate the opportunity to clarify this point.

As noted, the study employed a non-probabilistic convenience sampling approach. Recruitment was conducted primarily through digital channels, including institutional websites, social media platforms, and professional networks. More specifically:

• Institutional dissemination: An invitation to participate in the study was posted on official websites and internal bulletins of participating academic and healthcare institutions of Brasília, Federal District, Brazil.

• Social media: We utilized platforms such as Instagram and Facebook to circulate the study invitation. Posts included a brief description of the study objectives, eligibility criteria, and a phone number to schedule the first clinical study visit.

• Professional and academic groups: We shared the study invitation in closed messaging groups and online communities (e.g., WhatsApp groups) that included students, healthcare trainees, and professionals, particularly those involved in university hospitals or public health services.

• Snowball technique: We also encouraged participants and contacts to share the invitation with their peers, which further expanded the reach of our sample.

While we acknowledge the inherent limitations of this strategy, including the potential for selection bias and reduced generalizability, we adopted this approach to maximize reach during a period when in-person recruitment was not feasible due to the public health crisis context. To mitigate bias, we sought to diversify dissemination channels and target a broad range of participants from different regions, institutions, and backgrounds.

We have revised the manuscript to include these operational details in the Methods section and hope that this clarification adequately addresses your concern.

Criteria 4 - Conclusions are presented in an appropriate fashion and are supported by the data: Given the long study period and the known circulation of multiple SARS-CoV-2 variants in Brazil, the discussion should include a brief contextualization of the genomic landscape (e.g., VOCs/VOIs). Including this point will strengthen the interpretation of findings and increase clinical relevance, particularly for pregnant populations. Reference to appropriate literature (e.g., Male 2022) is encouraged.

Dear Editor,

Thank you very much for your careful reading and thoughtful suggestion regarding the inclusion of genomic surveillance and the circulation of SARS-CoV-2 variants of concern (VOCs) and interest (VOIs) in Brazil during the period of our study. We sincerely appreciate your perspective on how a brief discussion of this aspect could enhance the clinical relevance and contextual framing of our work, particularly regarding potential targeted interventions for pregnant women in the third trimester.

In response, we have incorporated a concise paragraph into the Discussion section addressing the dynamics of circulating SARS-CoV-2 variants in Brazil, referencing key genomic surveillance studies, as you suggested.

According to data from the European Centre for Disease Prevention and Control (ECDC) and the GISAID platform, the Alpha variant predominated from December 2020 to April 2021, being succeeded by Delta between June and December 2021, and later by Omicron, which became dominant from January 2022 [27,28].

Specifically, our study recruited patients between June 2020 and May 2021 — a period during which multiple SARS-CoV-2 lineages were circulating in Brazil. Early in this period, B.1.1.28 and B.1.1.33 lineages predominated; by late 2020, the VOI Zeta (P.2) emerged, followed by the VOC Gamma (P.1), first detected in Manaus, Brazil in November 2020 and becoming the dominant variant during the severe second wave in early 2021. In May 2021, the VOC Alpha (B.1.1.7) was also present, although with more limited circulation. These variants were characterized by distinct mutations, particularly in the spike protein, associated with increased transmissibility, immune escape, and, in some cases, higher clinical severity [29-31].

Variant epidemiology analyses highlight how genomic surveillance has been instrumental in tracking variant emergence, understanding transmission patterns, and informing public health interventions. For clinicians, particularly those caring for pregnant women in their third trimester, awareness of the predominant circulating variants is critical, as certain VOCs have been linked to increased risk of reinfection, and more severe disease presentations [29-31]. There is evidence that both maternal and neonatal outcomes were worse during the Delta wave of the SARS-CoV-2 pandemic than in preceding periods, but our study included mostly pregnant women with the Alpha or pre-Delta variant [32].

We hope this addition effectively addresses your suggestion and strengthens the manuscript’s impact. We thank you once again for your constructive feedback, which has helped improve the quality and clinical relevance of our work.

The reference list has been updated as follows:

27. EUROPEAN CENTRE FOR DISEASE PREVENTION AND CONTROL (ECDC). SARS-CoV-2 variants of concern as of [data]. Disponível em: https://www.ecdc.europa.eu/en/covid-19/variants-concern. Acesso em: 21 abr. 2025.

28. GISAID Initiative. Tracking of SARS-CoV-2 variants. Disponível em: https://www.gisaid.org/hcov19-variants/. Acesso em: 21 abr. 2025.

29. Vieira DFB, Bandeira DM, Nunes da Silva MA, de Almeida ALT, Araújo M, Machado AB, et al. Comparative analysis of SARS-CoV-2 variants Alpha (B.1.1.7), Gamma (P.1), Zeta (P.2) and Delta (B.1.617.2) in Vero-E6 cells: ultrastructural characterization of cytopathology and replication kinetics. Braz J Infect Dis. 2024;28(1):103706.

30. Resende PC, Naveca FG, Lins RD, Dezordi FZ, Ferraz MVF, Moreira EG, et al. Genomic epidemiology of the SARS-CoV-2 epidemic in Brazil. Nat Microbiol. 2022;7(9):1490–500.

31. Silva TS, Salvato RS, Gregianini TS, Gomes IA, Pereira EC, de Oliveira E, et al. Molecular characterization of a new SARS-CoV-2 recombinant cluster XAG identified in Brazil. Front Med (Lausanne). 2022;9:1008600.

32. Male V. SARS-CoV-2 infection and COVID-19 vaccination in pregnancy. Nat Rev Immunol. 2022 May;22(5):277-282.

Criteria 7 - The article adheres to appropriate reporting guidelines and community standards for data availability: The manuscript does not include a formal Data Availability Statement, as required by PLOS ONE policy. This is essential for ensuring transparency and reproducibility.

Dear Editor,

Thank you very much for your careful assessment and for highlighting the importance of data transparency and reproducibility.

We acknowledge that the initial submission did not include a formal Data Availability Statement, as required by PLOS ONE policy. We will address this by explicitly stating in the revised manuscript that:

All relevant data are available in the Harvard Dataverse repository: Sasaki, Lizandra, 2024, "Repli", https://doi.org/10.7910/DVN/GOHQUY.

This ensures that readers, reviewers, and the broader scientific community can fully access and evaluate the data underlying our findings, in line with the journal’s standards.

We sincerely appreciate your attention to this point, which helps us strengthen the rigor, reproducibility, and compliance of our work.

Reviewer's Responses to Questions

Comments to the Author

1. Is the manuscript technically sound, and do the data support the conclusions?

Reviewer #1: Yes; Reviewer #2: Yes

2. Has the statistical analysis been performed appropriately and rigorously?

Reviewer #1: Yes; Reviewer #2: Yes

3. Have the authors made all data underlying the findings in their manuscript fully available?

The PLOS Data policy requires authors to make all data underlying the findings described in their manuscript fully available without restriction, with rare exception (please refer to the Data Availability Statement in the manuscript PDF file). The data should be provided as part of the manuscript or its supporting information, or deposited to a public repository. For example, in addition to summary statistics, the data points behind means, medians and variance measures should be available. If there are restrictions on publicly sharing data - e.g. participant privacy or use of data from a third party - those must be specified.

Reviewer #1: No; Reviewer #2: Yes

4. Is the manuscript presented in an intelligible fashion and written in standard English?

Reviewer #1: Yes; Reviewer #2: Yes

5. Review Comments to the Author

Reviewer #1:

The manuscript clearly outlines a prospective cohort study in the methods, detailing the ethical approvals, study setting, and inclusion/exclusion criteria.

However, the recruitment strategy, while described as non-probabilistic convenience sampling, could benefit from a slightly more detailed explanation of how media and social networks were utilized.

The diagnostic criteria for SARS-CoV-2 infection, incorporating RT-PCR, serology, and chest CT findings as per Brazilian Ministry of Health guidelines, appear appropriate for the study period.

The Demographical and clinical records subsection describes the data collection process, including interviews, questionnaires, and follow-up schedules across pregnancy trimesters. The efforts to minimize response bias and follow-up losses are noted. The definitions of key variables, such as ethnicity, pregnancy trimester, COVID-19 phase, and severity (based on WHO criteria), are explicitly provided. The definitions for obstetric complications like GDM (IADPSG criteria), SAH, obesity, and fetal growth restriction are also clearly stated and reference relevant guidelines where applicable. Similarly, the definitions for premature birth and newborn classifications (INTERGROWTH-21st) enhance the rigor of the study.

The Statistical analysis outlines the methods used, including bivariate and multivariate analyses. The adjustment for pregestational control variables demonstrates an attempt to account for potential confounding factors.

The Results section systematically presents the findings, referencing specific tables. The description of labor complications and maternal mortality provides critical outcome data. The analysis of current and previous gestational diseases offers valuable clinical history.

However, the manuscript lacks a clear statement regarding the availability of the raw data underlying these findings. The PLOS ONE data policy requires this information to be explicitly stated, either with a link to a repository or a statement that the data are within the supplementary information. The absence of this (as the abstract and introduction do not mention it) is a significant concern that needs to be addressed to ensure the reproducibility and transparency of the research.

Reviewer #2:

Manuscript is well written and is covering all aspects already. I just have an optional suggestion that if genomic surveillance of variants could briefly be discussed? such as VOC/VOI because a several variants of SARS-CoV-2 were circulating in Brazil given the time span of the study so if it briefly describes will help clinicians for targeted intervention especially for the women in their third trimester. The thing I’m suggesting has been described by (Male 2022) so adding this aspect with respect to current study will enhance the impact of this study further.

Since all reviewer comments were included in the editor's request, we have addressed all questions in our responses to the editor.

We would like to once again thank the Editor and the Reviewers for their attention, support and guidance while revising our manuscript.

We are confident that the revised manuscript has been significantly improved by thoughtful inputs and hope that the present version is consistent with the high-quality articles of this journal and acceptable for publication in PLOS ONE.

Sincerely yours,

Lizandra Moura Paravidine Sasaki

Corresponding author on behalf of co-authors

---

## [Editor Report · Decision Letter 1]

Clinical characteristic and outcomes of pregnant women with COVID‐19: The PROUDEST prospective cohort study

PONE-D-25-15315R1

Dear Dr. Sasaki,

We’re pleased to inform you that your manuscript has been judged scientifically suitable for publication and will be formally accepted for publication once it meets all outstanding technical requirements.

Kind regards,

Elma Izze Da Silva Magalhães

Academic Editor

PLOS ONE

---

## [Editor Report · Acceptance letter]

PONE-D-25-15315R1

PLOS ONE

Dear Dr. Sasaki,

I'm pleased to inform you that your manuscript has been deemed suitable for publication in PLOS ONE. Congratulations! Your manuscript is now being handed over to our production team.

Kind regards,

on behalf of

Dr. Elma Izze Da Silva Magalhães

Academic Editor

PLOS ONE